# The NKG2D ligand ULBP4 is not expressed by human monocytes

**Mariya Lazarova[1], Younghoon Kim[1], Alexander Steinle[1,2]***

**1** Institute for Molecular Medicine, Goethe-University Frankfurt am Main, Frankfurt am Main, Germany,
**2** Frankfurt Cancer Institute, Frankfurt am Main, Germany

* alexander.steinle@kgu.de

**Data Availability Statement:** All relevant data are within the manuscript.

**Funding:** The authors received no specific funding for this work.

**Competing interests:** The authors have declared that no competing interests exist - except, of

## Abstract

The C-type lectin-like receptor NKG2D contributes to the immunosurveillance of virally infected and malignant cells by cytotoxic lymphocytes. A peculiar and puzzling feature of the NKG2D-based immunorecognition system is the high number of ligands for this single immunoreceptor. In humans, there are a total of eight NKG2D ligands (NKG2DL) comprising two members of the MIC (MICA, MICB) and six members of the ULBP family of glycoproteins (ULBP1 to ULBP6). While MICA has been extensively studied with regard to its biochemistry, cellular expression and function, very little is known about the NKG2DL ULBP4. This is, at least in part, due to its rather restricted expression by very few cell lines and tissues. Recently, constitutive ULBP4 expression by human monocytes was reported, questioning the view of tissue-restricted ULBP4 expression. Here, we scrutinized ULBP4 expression by human peripheral blood mononuclear cells and monocytes by analyzing ULBP4 transcripts and ULBP4 surface expression. In contrast to MICA, there was no ULBP4 expression detectable, neither by freshly isolated monocytes nor by PAMP-activated monocytes. However, a commercial antibody erroneously indicated surface ULBP4 on monocytes due to a non-ULBP4-specific binding activity, emphasizing the critical importance of validated reagents for life sciences. Collectively, our data show that ULBP4 is not expressed by monocytes, and likely also not by other peripheral blood immune cells, and therefore exhibits an expression pattern rather distinct from other human NKG2DL.

## Introduction

The human UL16-binding proteins (ULBPs) are a family of six MHC class I-related glycoproteins (ULBP1-6) encoded in the *RAET1* gene cluster of the long arm of human chromosome 6 [1,2]. They consist of an MHC class I-like α1α2 domain anchored in the cell membrane either by a GPI-anchor (ULBP1-3, ULBP6) or via a transmembrane domain (ULBP4, ULBP5) [1–4]. ULBPs together with the MHC-encoded MHC class I chain-related glycoproteins A and B (MICA and MICB) constitute the ligands of the activating immunoreceptor NKG2D [1–5].

NKG2D is a homodimeric C-type lectin-like receptor expressed on most human cytotoxic lymphocytes including NK cells, CD8 αβ T cells and γδ T cells [5,6]. NKG2D ligation not only promotes cytolysis of NKG2D ligand (NKG2DL)-expressing target cells but also cytokine

course, that we do research in the same research field.

secretion by the respective NKG2D[+] lymphocytes [5–8]. While NKG2DL are usually not expressed on healthy and quiescent cells, NKG2DL cell surfacing is facilitated by cellular stress, viral infection or malignant transformation, thus marking harmful cells for cytolysis [1,9–11]. Various viral immune evasion mechanisms target NKG2D-mediated recognition of virus-infected cells by suppressing NKG2DL expression and surfacing [12–15]. NKG2D also promotes cancer immunosurveillance and allows for detection of malignant cells subjected to genotoxic stress [10,16]. However, tumors obviously evolve mechanisms to escape from NKG2D-mediated immunosurveillance, including the downregulation of NKG2DL surface expression [17–20], proteolytic shedding of NKG2DL [21–24], and release of soluble NKG2DL in exosomes [25,26]. Hence, arming and exploiting NKG2D-mediated cancer surveillance appears promising for therapeutic targeting of cancer. First NKG2D-based cancer therapies are already in clinical trials [27–30]. However, a further in depth-characterization of the cellular and molecular basis of the NKG2D-NKG2DL interaction and of relevant tumor escape mechanisms is critical for the rational development of novel NKG2D-based immunotherapeutic approaches.

ULBP4 is a type I transmembrane glycoprotein of the ULBP family [31] and one of the least characterized NKG2DL. The lack of knowledge on ULBP4 is in part due to the scarcity of reliable detection reagents, since there are only a few "ULBP4-specific" antibodies commercially available with some even showing cross-reactivity to undefined antigen(s) on cells not containing ULBP4 transcripts [32]. ULBP4 is encoded by the *RAET1E* gene comprising five exons, which can give rise to several alternatively spliced transcripts and ULBP4 isoforms [2,32]. Unlike ULBP1 and ULBP2, ULBP4 is not bound by the name-giving HCMV glycoprotein UL16 [31,33], and therefore ULBP4 can be considered as a misnomer. The few studies on ULBP4 suggest that ULBP4 expression is highly restricted and distinct from other human NKG2DL [31,32]. ULBP4 transcripts were detected only in a few cell lines such as HeLa cells and a few tissues such as skin and esophagus, while detection of ULBP4 glycoproteins in these cells and tissues remains a challenge up to now [31,32]. Recently, a constitutive ULBP4 expression on human monocytes was reported [34] challenging the current view of highly restricted ULBP4 expression.

## Materials and methods

### PBMC isolation and monocyte enrichment

This study involving human blood samples from healthy volunteers has been approved by the local ethics committee of the Goethe-University Frankfurt am Main. Oral consent was obtained and documented. Blood samples were obtained from four healthy donors and PBMC isolated via density gradient centrifugation with Ficoll Paque Plus (GE Healthcare, Chicago, IL, USA). Monocytes were enriched by negative selection using a pan monocyte isolation kit (MiltenyiBiotec, Bergisch Gladbach, Germany) according to the manufacturer's protocol. This isolation kit allows the enrichment of all three major blood monocyte populations, i.e. "classical" (CD14[high]CD16[−]), "non-classical" (CD14[low]CD16[+]) and "intermediate" (CD14[high]CD16[+]) monocytes.

### Cell culture

Monocytes were cultured in RPMI 1640 medium (Sigma-Aldrich, St. Louis, MO, USA) supplemented with 10% fetal calf serum (FCS) (Biochrome, Berlin, Germany), 1 mM sodium pyruvate (Thermo Fisher, Waltham, MA, USA), 2 mM L-glutamine and 100 U/ml penicillin/100 mg/ml streptomycin (Sigma-Aldrich). For activation, media were supplemented with 500 ng/mL LPS (Sigma-Aldrich, St. Louis, MO, USA) or with 10 μg/mL poly(I:C) (Invitrogen,

Carlsbad, CA, USA). HeLa cells were cultured in Dulbecco's Modified Eagle Medium (Thermo Fisher Scientific, Waltham, MA, USA) supplemented with 10% FCS (Biochrome, Berlin, Germany), 2 mM L-glutamine and 100 U/ml penicillin/100 mg/ml streptomycin (Sigma-Aldrich).

### Antibodies

Anti-ULBP4 mAb (clone 709116) was purchased from R&D (Minneapolis, MN, USA). The ULBP4-specific mAb DUMO1 was previously described [32]. Anti-CD3-PerCP-Cy5.5 and fixable viability dye(FVD)-eFluor506 were from eBioscience (SanDiego, CA, USA), anti-CD19-APC-Cy7, anti-CD14-FITC, anti-NKp46-PE-Cy7, streptavidin-BV421 (SA-BV421) and goat-anti-mouse-IgG-PE (GaM-PE) were from Biolegend (SanDiego, CA, USA).

### Flow cytometry

Cells were first incubated with 30 μg/mL human IgG to block Fc-receptors for 20 min at 4˚C, and subsequently with the indicated mAb (10 μg/ml) for another 20 min at 4˚C. After washing, cell bound antibodies were stained with fluorochrome-conjugated goat anti-mouse IgG antibodies (Jackson ImmunoResearch). PBMC were subsequently stained with anti-CD3-PerCP-Cy5.5, anti-CD19-APC-Cy7, anti-CD14-FITC and anti-NKp46-PE-Cy7. Dead cells were excluded from the analysis by staining with fixable viability dye eFluor™ 506 (eBioscience). Flow cytometric analysis was performed on a FACS Canto II (BD Biosciences, Heidelberg, Germany) and data analyzed with FlowJo software (TreeStar, Ashland, Oh, USA).

### Quantitative RT-PCR (qRT-PCR)

RNA was isolated from pelleted cells with the RNAqueous Micro Total RNA isolation kit (Invitrogen) according to the manufacturer's protocol. Subsequently, RNA was treated with RNAse-free DNAse (Promega, Madison, WI, USA) and converted into cDNA using random primers (Promega) and M-MLV RT RNAse (H-) Point Mutant (Promega) according to the manufacturer's protocol. For quantification of ULBP4 and MICA transcripts, cDNA was amplified with ULBP4-specific (forward: 5'-ctg gct cag gga att ctt agg-3'; reverse: 5'-cta gaa gaa gac cag tgg ata tc-3') or MICA-specific primer pairs (forward: 5'-cct tgg cca tga acg tca gg-3'; reverse: 5'-cct ctg agg cct cac tgc g-3'). For normalization, 18S rRNA was detected (forward: 5'-cgg cta cca cat cca agg aa-3'; reverse: 5'-gct gga att acc gcg gct-3'). For positive control and normalisation, transcript levels of HeLa cells were determined. Quantitative PCR was performed with SYBR Green (Roche, Basel, Switzerland) on a QuantStudio 3 system (Applied Biosystems, Foster City, CA, USA). Relative gene expression normalized to 18S rRNA was calculated with the ΔΔCT method.

### Results

Recently, constitutive expression of ULBP4 on the cell surface of human monocytes was reported [34]. In order to assess these findings, we isolated peripheral blood mononuclear cells (PBMC) from the blood of four unrelated healthy donors and analyzed ULBP4 cell surface expression of the major leukocyte subpopulations by flow cytometry, including T cells (CD3[+]), B cells (CD19[+]), NK cells (CD3[-]CD19[-]NKp46[+]) and monocytes (CD3[-]CD19[-]CD14[+]) (Fig 1A and 1B). ULBP4 expression was assessed with the commercially available, allegedly "ULBP4-specific" monoclonal antibody (mAb) 709116 (R&D systems, clone 709116) utilized by Sharma et al., and with the ULBP4-specific mAb DUMO1, previously generated in our laboratory [32]. Both antibodies specifically bind to certain human cell lines only upon transfection with ULBP4 cDNA ("ULBP4 specificity") [32]. However, mAb 709116 in addition binds to a

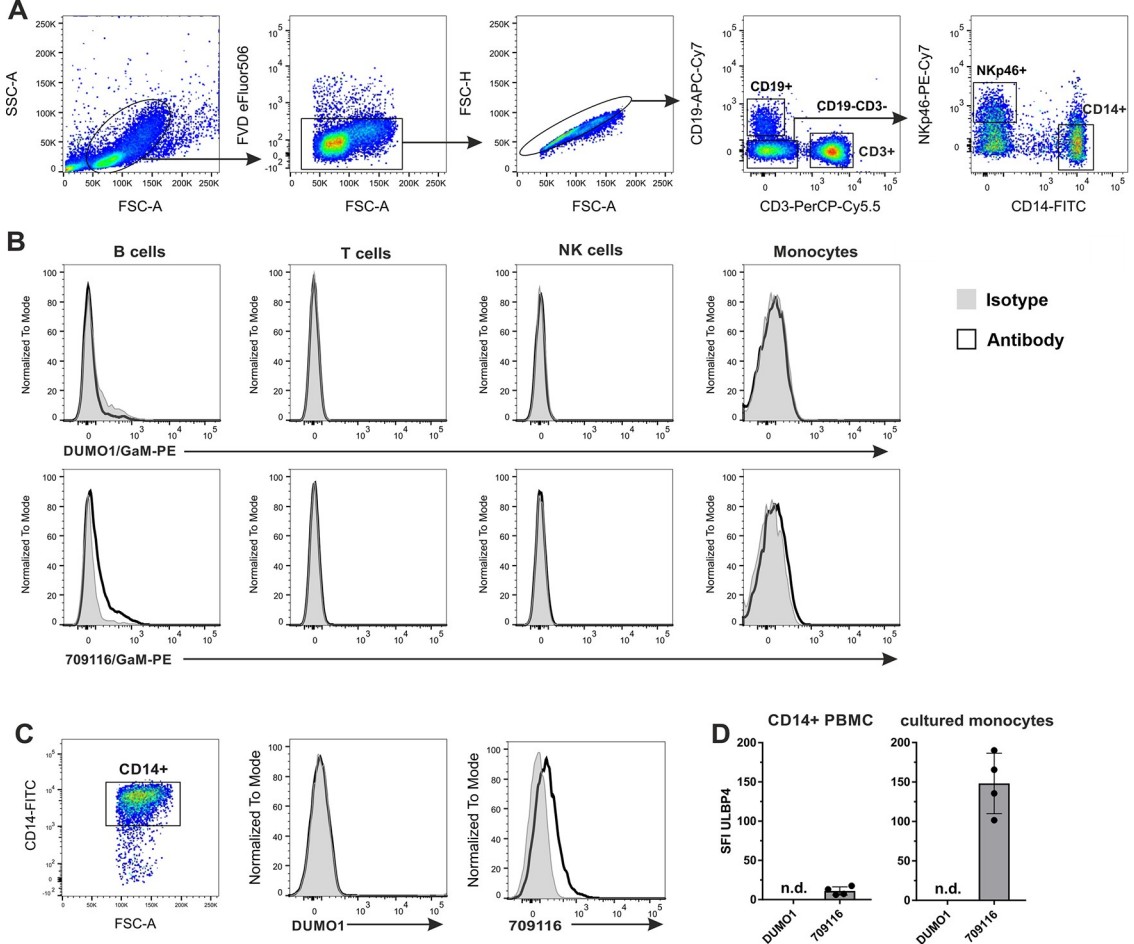

**Fig 1. Flow cytometric analysis of ULBP4 cell surface expression by human monocytes and other PBMC subsets.** (A, B) Freshly isolated human PBMC were analyzed by flow cytometry for ULBP4 surface expression using mAb DUMO1 or mAb 709116 with gates set on B cells (CD19+), T cells (CD3+), NK cells (CD3−CD14−CD19−NKp46+) and monocytes (CD3−CD14+CD19−NKp46−), respectively. (A) Representative gating strategy is shown (B) Representative histogram overlays of ULBP4 stainings (solid line) and isotype controls (gray filled). (C) Monocytes enriched from PBMC were cultured for 16 h and subsequently analyzed by flow cytometry for ULBP4 surface expression with DUMO1 or 709116. Representative gating of cultured CD14+ cells (left) and overlays of DUMO1 or 709116 stainings (solid lines), respectively, with isotype controls (gray). (D) Comprehensive depiction of ULBP4 surface expression on either monocytes (CD3−CD14+CD19−NKp46−) among freshly isolated PBMC (left) or on enriched CD14+ monocytes after 16 h culture (right) of four human healthy donors. Specific fluorescence intensity (SFI) is shown (SFI = MFI (ULBP4 mAb)—MFI (isotype control)). Bars represent means with SD indicated; n. d. = not detectable.

human cell line lacking ULBP4 transcripts thus binding to another yet unknown antigen in a cross-reactive manner [32]. We detected a marginal binding of mAb 709116 (as compared to the isotype control) to B cells, but not to any of the other subsets of all tested donors (Fig 1B). Similarly, no DUMO1 binding was detected to any of the immune cell subsets of any of the donors. These results suggested that there are no ULBP4 molecules on the cell surface of T cells, NK cells, and monocytes. To test whether ULBP4 surface expression by monocytes may be inducible upon activation, as reported for other human NKG2DL [35], we broadly enriched monocytes from freshly isolated PBMC using a pan monocyte isolation kit (S1 Fig), and cultured these *in vitro* for 16 h prior to flow cytometric analysis. Again, staining with DUMO1 did not detect any ULBP4 expression on cultured CD14+ monocytes (Fig 1C and 1D). In contrast, mAb 709116 markedly bound to cultured CD14+ monocytes of all four donors tested,

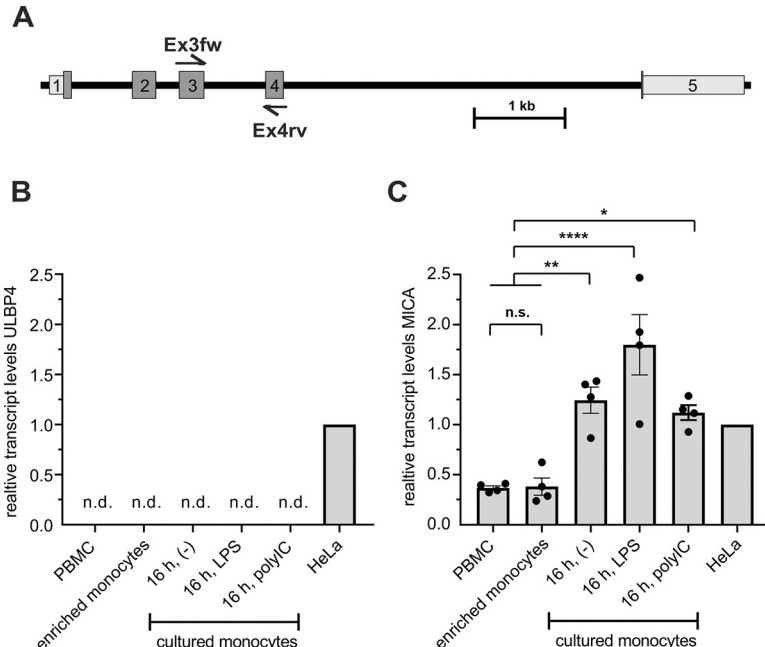

**Fig 2. No ULBP4 transcripts detectable in human PBMC and monocytes.** (A) Schematic localization of PCR primer sites (Ex3fw, Ex4rv) of the ULBP4-specific qRT-PCR within in the ULBP4-encoding *RAET1E* gene locus. The primer pair was designed to detect all known ULBP4 isoforms transcripts. (B) No ULBP4 transcripts detectable by qRT-PCR in PBMC-derived samples of four human donors including freshly isolated PBMC, broadly enriched monocytes, or broadly enriched monocytes after 16 h culture in the absence of a stimulus (-), or in the presence of lipopolysaccharide (LPS) (at 500 ng/mL) or polyinosinic:polycytidylic acid (poly(I:C)(at 10 μg/mL), respectively. HeLa cells were used as positive control. (C) For quality control, all samples were also analyzed by qRT-PCR for MICA transcripts. (B, C) Expression levels were normalized to 18S rRNA and set relative to the transcript levels of HeLa cells. Bars represent means with SEM indicated; n. d. = not detectable. Statistical analysis was performed by one-way Anova (* $p < 0.05$, ** $p \leq 0.005$, **** $p < 0.0001$; n.s.–not significant).

similarly to the results reported by Sharma et al (Fig 1C and 1D). We further stimulated enriched monocytes with the TLR4 ligand lipopolysaccharide (LPS; at 500 ng/mL) or with the TLR3 ligand polyinosinic:polycytidylic acid (poly(I:C); at 10 μg/mL) for 16 h in culture before analysis of ULBP4 surface expression. However, no DUMO1 binding to any of the stimulated monocytes was observed (S2 Fig).

Because of the discrepancy between the stainings of DUMO1 and mAb 709116, we addressed ULBP4 expression at the transcript level for both PBMC and enriched monocytes. To this aim, we performed qRT-PCR using primers with binding sites in exon 3 and exon 4, respectively, which allow detection of all functional cell-bound ULBP4 isoforms (Fig 2A). We performed qRT-PCR for ULBP4 with cDNA from freshly isolated PBMC and from monocytes enriched from PBMC, either directly after enrichment, or after 16 h of *in vitro* cultivation either in the absence (unstimulated) or in the presence of LPS (500 ng/mL) or poly(I:C) (10 μg/mL), respectively. The human cervix carcinoma cell line HeLa, previously shown to express ULBP4 [32], was used as a positive control. While ULBP4 transcripts were present in HeLa cells at substantial levels, no ULBP4 transcripts were detectable, neither in whole PBMC nor in any of the freshly enriched or cultured or TLR-stimulated monocytes (Fig 2B). For quality control of the experimental process, we additionally determined the expression of the NKG2DL MICA by qRT-PCR (Fig 2C). MICA transcripts were clearly detectable in all samples as expected from previous reports [35]. For an assessment of these data, it is important to add that ULBP4 transcript levels of HeLa cells do not suffice for detectable surface ULBP4

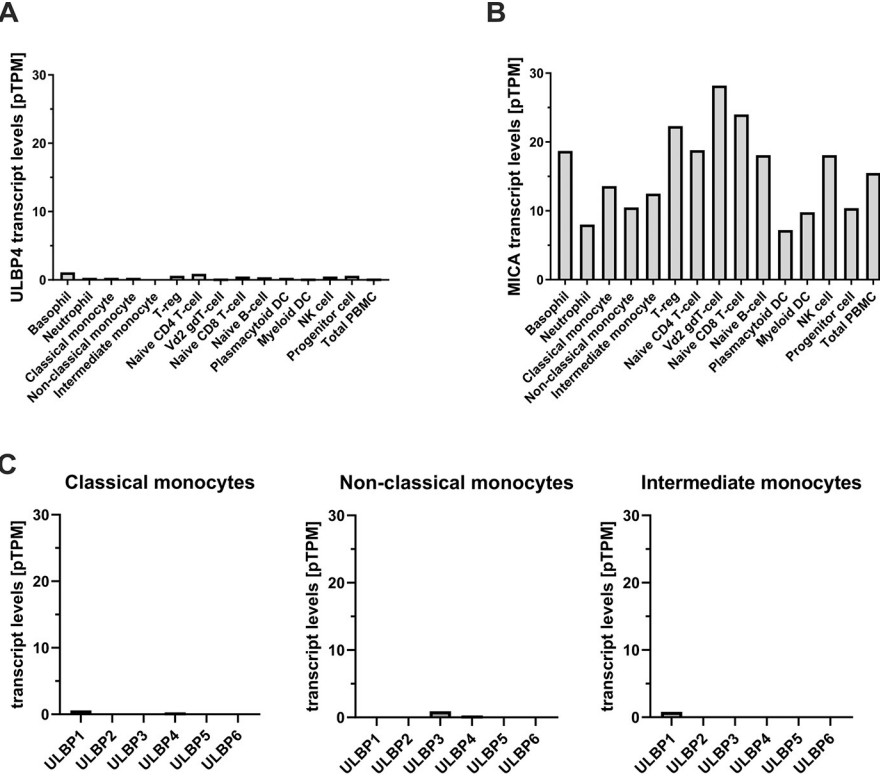

**Fig 3. Mining of RNASeq datasets for human PBMC subsets reveal no or minimal ULBP4 expression.** (A) Number of ULBP4 transcripts (pTPM: transcripts per million) and (B), for comparison, of MICA transcripts in several immune cell subsets, including various monocyte subsets. (C) Number of transcripts for all six ULBP family members in the three major blood monocyte populations (pTPM: transcripts per million). Data were reported by Monaco and colleagues [37] and retrieved via the human protein atlas (https://www.proteinatlas.org).

expression [32] and that DUMO1 and mAb 709116 roughly bind with the same "strength" to ULBP4-transfected cells [32].

Finally, we mined publicly available databases for ULBP4 transcripts in human blood cell populations [36–38]. Schmiedel and colleagues analysed the RNA contents of 13 subsets of human immune cells derived from leukapheresis samples of 91 healthy donors by RNASeq. These data are publicly available in the database of immune cell expression (DICE): https://dice-database.org. In this dataset, no ULBP4 transcripts were detected in any blood cell subset except in memory CD4 T cells where extremely low levels (0.1 transcripts per million, (TPM)) were found [36] (see https://dice-database.org/genes/raet1e). Monaco and colleagues provide data for 29 different types of immune cells, including various subsets of CD4 and CD8 T cells, unconventional T cells, B cells, NK cells, monocytes, dendritic cells, granulocytes and immune cell progenitors [36–38]. In this dataset, extremely low levels of ULBP4 transcripts (ranging from 0.2 to 1.1 pTPM) are reported for the analyzed immune cell subsets. Data of ULBP4 transcripts (Fig 3A), and for control of MICA transcripts (Fig 3B), of the main immune cell populations from this published dataset are represented in Fig 3. Of note, according this publicly available dataset [36–38], monocytes not or very poorly express ULBP family members in general (Fig 3C). A third study within the framework of the human protein atlas project recently reported the transcriptome analysis for 18 immune cell subsets derived from six healthy donors [38]. Here, ULBP4 transcripts were detected at extremely low levels in basophils, neutrophils, monocytes, T cell subsets, B cells and NK cells (0.1 to 0.4 pTPM). All datasets

concordantly show that ULBP4 transcripts are either absent or present at extremely low levels in blood cells in general, and in monocytes, in particular. Considering that cells have been estimated to contain between 50.000 and 300.000 mRNA molecules [39], an abundance of 0.1 to 1 TPM as determined for ULBP4 in various immune subsets would accordingly correspond to one ULBP4 transcript per 3–180 cells. Obviously, such expression would neither result in detectable surface expression nor translate into biological significance for a given immune cell subset.

## Concluding remarks

The validity of experimental data strongly depends on the quality and specificity of the reagents used. Here we report on another unfortunate example how a cross-reactive antibody does cause misinterpretation of experimental data. The commercially available "ULBP4 specific" mAb 709116 binds to human monocytes (our data and [34]). However, both the qRT-PCR data presented here and datasets of three independent databases show that there are virtually no ULBP4 transcripts in human monocytes. Lack of ULBP4 surface expression by monocytes was also confirmed by the validated ULBP4-specific mAb DUMO1. Obviously, mAb 709116 cross-reacts with a surface molecule on human cells, which is not ULBP4, as already stated previously [32]. This emphasizes that experimental data obtained with the cross-reactive mAb 709116 antibody must be interpreted with caution and require validation by independent methods. Collectively, the presented data exclude that ULBP4 is expressed by human monocytes under normal conditions and upon activation by PAMP. In addition, our data together with publicly available data sets also strongly suggest that ULBP4 is not expressed by peripheral blood mononuclear cells in contrast to other human NKG2DL. These data support the notion of ULBP4 fulfilling a non-redundant, tissue-specific function within the human immune system that awaits elucidation by future studies.

## Supporting information

**S1 Fig. Enrichment of monocytes.** Monocytes enriched from PBMC using a pan monocyte isolation kit were directly analysed by flow cytometry to validate the enrichment. All three major blood monocyte populations (classical (CD14$^{high}$CD16$^-$), non-classical (CD14$^{low}$CD16$^+$) and intermediate (CD14$^{high}$CD16$^+$) monocytes) were successfully enriched. The percentage of total monocytes is indicated.
(PDF)

**S2 Fig. DUMO1 does not bind TLR-stimulated monocytes.** Monocytes enriched from PBMC were cultured for 16 h in the presence of lipopolysaccharide (LPS) (at 500 ng/mL) or with polyinosinic:polycytidylic acid (poly(I:C) (at 10 μg/mL) and subsequently analyzed by flow cytometry for ULBP4 surface expression with DUMO1. Overlays of DUMO1 stainings (solid lines) with isotype controls (gray filled) of CD14$^+$ monocytes are shown for all four donors. Note that isotype controls, but not DUMO1, unspecifically stained a minor portion of cells.
(PDF)

## Acknowledgments

We would like to thank the blood donors and Christina Born, Nikita Verheyden, and Ines Kühnel for technical support and advice.

## Author Contributions

**Conceptualization:** Alexander Steinle.

**Data curation:** Mariya Lazarova.

**Formal analysis:** Alexander Steinle.

**Funding acquisition:** Alexander Steinle.

**Investigation:** Mariya Lazarova, Younghoon Kim.

**Methodology:** Mariya Lazarova, Younghoon Kim.

**Supervision:** Alexander Steinle.

**Validation:** Alexander Steinle.

**Writing – original draft:** Mariya Lazarova.

**Writing – review & editing:** Alexander Steinle.

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
