## [Decision Letter · Decision Letter 0]

21 May 2020

PONE-D-20-08786

The NKG2D ligand ULBP4 is not expressed by human monocytes

PLOS ONE

Dear Dr. Steinle,

Thank you for submitting your manuscript to PLOS ONE. After careful consideration, we feel that it has merit but does not fully meet PLOS ONE’s publication criteria as it currently stands. Therefore, we invite you to submit a revised version of the manuscript that addresses the points raised by the reviewers during the review process.

We would appreciate receiving your revised manuscript by Jul 05 2020 11:59PM. To enhance the reproducibility of your results, we recommend that if applicable you deposit your laboratory protocols in protocols.io, where a protocol can be assigned its own identifier (DOI) such that it can be cited independently in the future. For instructions see: http://journals.plos.org/plosone/s/submission-guidelines#loc-laboratory-protocols

We look forward to receiving your revised manuscript.

Kind regards,

Silke Appel, PhD (Dr. rer. nat.)

Academic Editor

PLOS ONE

Journal Requirements:

2. Please amend your current ethics statement to address the following concerns: Please explain why written consent was not obtained, how you recorded/documented participant consent, and if the ethics committees/IRBs approved this consent procedure

3. Thank you for including your ethics statement in your methods section:

"Blood samples were obtained from four healthy donors with approval of the local Ethics

Committee"

Once you have amended this statement in the Methods section of the manuscript, please add the same text to the “Ethics Statement” field of the submission form (via “Edit Submission”).

Reviewers' comments:

Reviewer's Responses to Questions

**Comments to the Author**

1. Is the manuscript technically sound, and do the data support the conclusions?

Reviewer #1: Yes

Reviewer #2: Partly

Reviewer #3: Partly

2. Has the statistical analysis been performed appropriately and rigorously? 

Reviewer #1: N/A

Reviewer #2: N/A

Reviewer #3: N/A

3. Have the authors made all data underlying the findings in their manuscript fully available?

Reviewer #1: No

Reviewer #2: Yes

Reviewer #3: Yes

4. Is the manuscript presented in an intelligible fashion and written in standard English?

Reviewer #1: Yes

Reviewer #2: Yes

Reviewer #3: Yes

5. Review Comments to the Author

Reviewer #1: I have suggested minor revision as I feel the publication satifys the criteria for publication in PLOS ONE

The paper is well written, easily understood, concise and provides a good search of the literature. The paper highlights the need to properly test and validate antibodies even from commercial retailers to avoid incorrect conclusions. This you have successfully highlighted in Sharma and Markiewicz 2019. This highlights the need for additional testing not reliant on specificity of an antibody, in particular when results are not as expected as in the case of Sharma and Markiewicz 2019. The examination of both ULBP4 transcript levels in addition to the use of an in-house produced monoclonal antibody and the use of a positive control further re enforces the validity of your finding that monocytes do not express ULBP4, in contrast to the findings of Sharma and Markiewicz 2019. Further the investigation of 3 different databases in regards to ULBP4 transcripts in immune cells further supports your conclusion.

The minor revisions are in regards to the presented figure.

1) Figure 1A, part of the flow cytometry gating strategy is missing- in particular the initial cell gate and the live dead stain.

2) Figure 1 and 2 ideally should be edited to be lighter, this is in particular regards to Figure 2C where datapoints are difficult to visualize.

3) Figure 1B shows a change in the color of the presented histograms (blue, all except one) than what is described in the text (gray). This should be corrected.

4) Figures 1D and 2C need to state what the bars represent (means or medians, IQR?)

5) Figure 1D and 2B should state what n,d respresents (assuming not detected) in figure text.

6) Flow cytometry data should be uploaded to the flow respository or similar to be made publically avaliable.

Reviewer #2: The manuscript from Lazarova and colleagues reports the lack of expression of the NKG2D ligand ULBP4 in the major immune cell subsets within the PBMC fraction, at least in the tested conditions. They highlight the divergent results obtained with two antibody clones reported to bind ULBP4, and suggested one of them to be non-specific. In general, I consider reporting on negative results a very important issue, especially in a period of “reproducibility crisis”. Therefore, I would recommend this paper for publication, upon revision of some minor points, listed below.

- A major limitation of this study is the lack of data on the CD14- monocyte fraction. This applies to every figure provided, and is particularly relevant as the current title refers to (total) “human monocytes”. In order to support with data their general statement, I would here recommend to perform experiments on the total monocyte population. Another reason for this is that, as the authors report in Figure 3B, different monocyte subsets can express different levels of NKG2D ligands, at least for what concerns MICA RNA. Where applicable, it would be useful to know whether ULBP4 expression changes across subsets.

-A related point is on the qPCR experiments (Figure 2B), where, again, there are no stimulated CD16+ CD14- monocytes. Also, since they perform qPCR on an “enriched” cellular fraction, the authors should provide data on the purity of isolated cells. Otherwise, FACS-sorting would be appreciated.

- Figure 1: A relevant point here is the lack of any type of positive control for the DUMO1 antibody binding (even on non-immune cells/on cell lines/transfectants). If possible I think the authors should provide it. As a minor technical comment, the staining for NK cells is suboptimal, and it would be nice to see a quantification on CD56+ CD3- cells.

- The use of public RNAseq datasets is convincing and the MICA vs MICB difference in absolute counts is striking. To complement these data, I would encourage the authors to include data relative not only to steady state conditions, but also to inflammatory conditions where, theoretically, ULBP4 might be involved (e.g. cancer). There is a number of repositories human of bulk/single cell sequencing data from immune/stromal cells where this information can easily be collected. This would nicely pair up with the in vitro stimulation data presented in Figure 2.

- Minor text point:

Page 4, line 12 “which are in part not ULBP4-specific”, can it be made clearer?

Reviewer #3: In the current study by Lazarova M. et al the authors report the unspecific binding of the anti-ULBP4 mAb 709116 (R&D systems).The effort made by the author to advise the scientific community about the careful usage of such reagents is considerable. However their statements would be better supported by showing unspecific binding of the mAb 709116 in a cell line that do not express the endogenous protein and that is transfected with a plasmid encoding ULBP4 or a control vector. Finally it will be interesting to understand in the future the specific role and tissue-restricted expression of ULBP4 compared to the other members of the same family.

Comments for figure 1:

□ The authors should also include the forward and side scatter plot to show proper gating of monocytes and lymphocytes that are characterized by different size

□ In figure 1B the author should correct the color on the histogram for the monocytes stained with the 709116 ab to much the other histograms

□ In figure 1B and C the authors should include legends to help undertanding which is the isotype and which is the stained sample

□ The authors should specify the stimulation conditions for the monocytes in the text, in the figure and in the figure legend

□ Is the 709116 ab detecting ULBP4 expression also after TLR3 and TLR4 stimulation?

Comments for figure 3:

□ The authors should include expression levels of other proteins (ULBPs) of the same family from the same dataset

□ The authors should include proteomic data from the ProteinAtlas dataset to corroborate their conclusions

Statistical analysis must be provided

6. PLOS authors have the option to publish the peer review history of their article (what does this mean?). If published, this will include your full peer review and any attached files.

Reviewer #1: No

Reviewer #2: Yes: Andrea Ponzetta

Reviewer #3: No

---

## [Author Response · Author response to Decision Letter 0]

12 Jan 2021

Response to Reviewers

Dear Dr. Appel,

We greatly appreciate your kind consideration of our manuscript for publication in PLOS ONE and thank you and the Reviewers for the positive responses to our manuscript.

Please find below and as a separate file attached to the manuscript our point-to-point reply to all comments raised:

Editorial comments:

Reply: We adjusted our manuscript to the style requirements of PLOS ONE. 

2. Please amend your current ethics statement to address the following concerns: Please explain why written consent was not obtained, how you recorded/documented participant consent, and if the ethics committees/IRBs approved this consent procedure

Reply: Blood samples were obtained with the approval of the local Ethics Committee of the University Hospital Frankfurt. We analyzed peripheral blood leukocytes of 4 healthy human donors using ~20 - 40 ml peripheral blood drawn from the arm vein. All 4 donors are/were employed scientists at our Institute for Molecular Medicine, with one of them (YK) also being co-author of this manuscript. All 4 donors volunteered to donate blood upon a general request for blood donation and upon describing the scientific purpose, i.e. the analysis of NKG2DL expression of peripheral blood cells - a purpose that was fully understood as a consequence of the scientific education and membership of the Institute. All 4 donors explicitly consented orally to donate blood for this specific purpose which was documented. All donors volunteered to donate blood in order to support the advance of science in this specific research area without financial compensation.

3. Thank you for including your ethics statement in your methods section:

"Blood samples were obtained from four healthy donors with approval of the local Ethics

Committee"

Once you have amended this statement in the Methods section of the manuscript, please add the same text to the “Ethics Statement” field of the submission form (via “Edit Submission”).

Reply: We amended the text on page 4 as follows: 

"Blood samples were obtained from four healthy donors with the approval of the local Ethics Committee of the University Hospital Frankfurt." 

Reply: We included the corresponding data (DUMO1 does not bind TLR- stimulated monocytes) now as Supporting Figure S2. 

5. Review Comments to the Author

Reviewer #1: I have suggested minor revision as I feel the publication satifys the criteria for publication in PLOS ONE

The paper is well written, easily understood, concise and provides a good search of the literature. The paper highlights the need to properly test and validate antibodies even from commercial retailers to avoid incorrect conclusions. This you have successfully highlighted in Sharma and Markiewicz 2019. This highlights the need for additional testing not reliant on specificity of an antibody, in particular when results are not as expected as in the case of Sharma and Markiewicz 2019. The examination of both ULBP4 transcript levels in addition to the use of an in-house produced monoclonal antibody and the use of a positive control further re enforces the validity of your finding that monocytes do not express ULBP4, in contrast to the findings of Sharma and Markiewicz 2019. Further the investigation of 3 different databases in regards to ULBP4 transcripts in immune cells further supports your conclusion.

Reply: We appreciate the positive comments of R1 regarding our study.

The minor revisions are in regards to the presented figure.

1) Figure 1A, part of the flow cytometry gating strategy is missing- in particular the initial cell gate and the live dead stain.

Reply: We now extended Fig. 1A by the full requested gating strategy.

2) Figure 1 and 2 ideally should be edited to be lighter, this is in particular regards to Figure 2C where datapoints are difficult to visualize.

Reply: We now include new versions of Figure 1D and Figure 2 where bars are given in lighter gray.

3) Figure 1B shows a change in the color of the presented histograms (blue, all except one) than what is described in the text (gray). This should be corrected.

Reply: We thank R1 for alerting us.We now changed all histograms to same color (i.e. gray). 

4) Figures 1D and 2C need to state what the bars represent (means or medians, IQR?)

Reply: We do apologise having missed out. We now included these informations in the respective figure legends (Fig. 1D and 2C).

5) Figure 1D and 2B should state what n,d respresents (assuming not detected) in figure text.

Reply: We now define n.d. as "not detectable" in the respective figure legends.

6) Flow cytometry data should be uploaded to the flow respository or similar to be made publically avaliable.

Reply: We now show the additional flow cytometric data in the Supplement Figures.

Reviewer #2:

 The manuscript from Lazarova and colleagues reports the lack of expression of the NKG2D ligand ULBP4 in the major immune cell subsets within the PBMC fraction, at least in the tested conditions. They highlight the divergent results obtained with two antibody clones reported to bind ULBP4, and suggested one of them to be non-specific. In general, I consider reporting on negative results a very important issue, especially in a period of “reproducibility crisis”. Therefore, I would recommend this paper for publication, upon revision of some minor points, listed below.

Reply: We thank R2 for the appreciation of our work.

- A major limitation of this study is the lack of data on the CD14- monocyte fraction. This applies to every figure provided, and is particularly relevant as the current title refers to (total) “human monocytes”. In order to support with data their general statement, I would here recommend to perform experiments on the total monocyte population. Another reason for this is that, as the authors report in Figure 3B, different monocyte subsets can express different levels of NKG2D ligands, at least for what concerns MICA RNA. Where applicable, it would be useful to know whether ULBP4 expression changes across subsets.

Reply: We are indebted to R2 for raising this point. In fact, we did the experiments shown in Figure 2 on the total monocyte population as recommended by R2. For qPCR analyses, monocytes were enriched up to 91% purity using the pan monocyte purification kit by Miltenyi, which enriches all three major types of blood monocytes by negative selection, i.e. classical CD14highCD16- monocytes, non-classical CD14lowCD16+ monocytes, and intermediate CD14highCD16+ monocytes as can be seen in our new Supplement Figure S1. Since we did not detect any ULBP4 transcripts in the enriched bulk of these monocytes by qPCR (Fig 2B), we also can exclude that any of the three subsets comprised herein is expressing ULBP4 at biologically relevant levels.

To make this clearer, we now specified our manuscript throughout accordingly to more specifically represent our data. We also now introduce the three described major monocyte populations, i.e. classical CD14+CD16- monocytes, non-classical CD14lowCD16+ monocytes, and intermediate CD14+CD16+ monocytes for the reader. We also specify for all data sets which of monocytes or monocyte subsets were analysed. 

We are grateful to R2 for pointing this out, which helped us to substantially improve our manuscript.

In addition, we would like to state the following:

1) The main and major point of this manuscript is to alert the scientific community to our findings that the recent report of "constitutive ULBP4 expression by human monocytes" by Sharma and Markiewicz is not correct. Sharma and Markiewicz analyzed classical CD14+ monocytes (as most researchers do) by flow cytometry and therefore our focus is/was also on this major subset, e.g. in the flow cytometric studies shown in Figure 1, enabling a direct comparison of results.

2) The transcriptomes of the three major monocyte subsets were separately analyzed by RNAseq in the studies of Schmiedel et al, Monaco et al., and Uhlen et al., respectively. In all the datasets from all three studies there are either no ULBP4 (RAET1E) transcripts at all in any of the three subsets (Schmiedel et al., ) or near-to-nothing levels (Monaco et al., see Fig. 3A)(Uhlen et al., ). In our study, we analyzed bulk monocytes comprising all three major subsets not detecting any ULBP4 transcripts corroborating the RNAseq data.

Collectively, these data sets unambiguously and clearly show that there is no relevant ULBP4 (RAET1E) expression in any of the three major monocytes subsets of human peripheral blood.

-A related point is on the qPCR experiments (Figure 2B), where, again, there are no stimulated CD16+ CD14- monocytes. Also, since they perform qPCR on an “enriched” cellular fraction, the authors should provide data on the purity of isolated cells. Otherwise, FACS-sorting would be appreciated.

Reply: As mentioned above, the pan monocyte selection kit by Miltenyi does also enrich for CD16+CD14- monocytes, which consequently are included in the resulting data (i.e. no ULBP4 transcripts detectable). We apologise for not having made this clearer in our original version of the manuscript, which we now amended accordingly. We now also show the purity of the MACS-enriched monocytes in the new Supplement Figure S1.

- Figure 1: A relevant point here is the lack of any type of positive control for the DUMO1 antibody binding (even on non-immune cells/on cell lines/transfectants). If possible I think the authors should provide it. 

Reply: We extensively characterized DUMO1 (including many positive and negative controls) in our original publication of DUMO1 (Zöller et al., 2018). We did not include these data again in the current manuscript because this may have represented a "duplicate" publication.

As a minor technical comment, the staining for NK cells is suboptimal, and it would be nice to see a quantification on CD56+ CD3- cells.

Reply: We respectfully disagree here. while gating on CD3-NKp46+ cells for NK cells is state of the art, a quantification of gated NK cells in the context of our study is not of major relevance.

- The use of public RNAseq datasets is convincing and the MICA vs MICB difference in absolute counts is striking. To complement these data, I would encourage the authors to include data relative not only to steady state conditions, but also to inflammatory conditions where, theoretically, ULBP4 might be involved (e.g. cancer). There is a number of repositories human of bulk/single cell sequencing data from immune/stromal cells where this information can easily be collected. This would nicely pair up with the in vitro stimulation data presented in Figure 2.

Reply: We assume that R2 mistakenly refers to a MICA vs. MICB difference, but rather intends to refer to a MICA vs ULBP4 difference? We do agree that it will be very interesting to thoroughly address expression of all 8 human NKG2DL by immune cells in various inflammatory or autoimmune or cancerous environments, but these extensive analyses are far beyond the present scope of our manuscript, which is a contradictory report to a recent study claiming constitutive ULBP4 expression by human blood monocytes".

- Minor text point:

Page 4, line 12 “which are in part not ULBP4-specific”, can it be made clearer?

Reply: For clarification, we amended the text as follows: 

“The lack of knowledge on ULBP4 is in part due to the scarcity of reliable detection reagents, since there are only a few "ULBP4-specific" antibodies commercially available with some even showing cross-reactivity to undefined antigen(s) on cells not containing ULBP4 transcripts (Zöller et al., 2018).”

Reviewer #3: 

In the current study by Lazarova M. et al the authors report the unspecific binding of the anti-ULBP4 mAb 709116 (R&D systems).The effort made by the author to advise the scientific community about the careful usage of such reagents is considerable.

Reply: We thank R3 for the appreciation of our study.

 However their statements would be better supported by showing unspecific binding of the mAb 709116 in a cell line that do not express the endogenous protein and that is transfected with a plasmid encoding ULBP4 or a control vector. 

Reply: In our previous study on ULBP4 (Zöller et al., 2018), we had provided clear evidence both (1) for unspecific (= non-ULBP4-specific) binding of mAb 709116 to a cell line and (2) for ULBP4-specific binding of mAb 709116 to an ULBP4-negative cell line upon transfection with an ULBP4 cDNA.

(1) we reported in Figure 4 that mAb 709116 unspecifically (non-ULBP4-specifically) binds to human HepG2 cells (hepatocellular carcinoma cell line), since HepG2 cells do not contain any ULBP4/RAET1E transcripts (similarly to monocytes as shown in our present manuscript). 

(2) In that previous study, we also provide clear evidence in Figure 3 that human C1R cells (B cell line) are only bound by mAb 709116 (similarly to DUMO1) upon transfection with ULBP4 cDNA, whereas mock-transfected C1R cells are not bound by mAb 709116.

Hence, mAb709116 binds to some human cells (e.g. monocytes, HepG2) in an ULBP4-independent manner (since these cells lack ULBP4 transcripts), whereas it binds to other human cells such as C1R cells only upon ULBP4 transfection. These previously published data demonstrate that mAb709116 binds to at least two different human antigens, namely, to a) ULBP4 and to b) - in a cross-reactive manner - to an unknown antigen present on HepG2 cells and human monocytes.

In contrast, our self-made mAb DUMO1 specifically binds to ULBP4-transfected cell lines (but not mock-transfected cells) and recombinantly produced ULBP4, but does not bind to cells that do NOT contain ULBP4/RAET1E transcripts such as HepG2 cells or human monocytes (see Zöller et al., 2018 and our present manuscript). 

Taken together, this point addressing mAb 709116 specificity raised by R3 has already been addressed in our previous study, and representing these data here once again would represent a "duplicate" publication.

Finally it will be interesting to understand in the future the specific role and tissue-restricted expression of ULBP4 compared to the other members of the same family.

Reply: We fully agree with R3 and are actively researching on this challenging research topic with some preliminary observations that are far from being published at present.

Comments for figure 1:

The authors should also include the forward and side scatter plot to show proper gating of monocytes and lymphocytes that are characterized by different size

Reply: As already suggested by R1 we now show the entire gating strategy in Fig. 1A.

In figure 1B the author should correct the color on the histogram for the monocytes stained with the 709116 ab to much the other histograms

Reply: We thank R3 for pointing out this error which was corrected as already mentioned in our reply to R1.

In figure 1B and C the authors should include legends to help undertanding which is the isotype and which is the stained sample

Reply: We included such legends in figure 1 itself in addition to the explanations in the figure legend.

The authors should specify the stimulation conditions for the monocytes in the text, in the figure and in the figure legend

Reply: We now specified the conditions in the text, in addition to the figure (Fig 2) and the Figure legend.

Is the 709116 ab detecting ULBP4 expression also after TLR3 and TLR4 stimulation?

Reply: We did not analyze mAb 709116 binding after TLR3 or TLR4 stimulation for the following reasons:

(1) Sharma and Markiewicz claimed "constitutive" ULBP4 expression on non-stimulated monocytes. Hence, the primary aim of this study is to inform the scientific community that this observation using non-stimulated monocytes is not correct. 

(2) In addition, we also assessed ULBP4/RAET1E transcripts of monocytes treated with the TLR3 and TLR4 ligands polyI:C and LPS, respectively (Figure 2) and could not detect any ULBP4 transcripts by highly sensitive qPCR, excluding that ULBP4 is up-regulated on monocytes upon stimulation with TLR ligands, which is the relevant information. In line with these results, there was also no surface ULBP4 detectable with mAb DUMO1 upon treatment with LPS or poly(I:C) (S2 Fig) demonstrating that there is also no ULBP4 on TLR3/TLR4 stimulated monocytes.

Comments for figure 3:

The authors should include expression levels of other proteins (ULBPs) of the same family from the same dataset

Reply: We thank R3 for this suggestion and therefore added an additional Figure 3C where expression levels (transcripts) of all six ULBP family members (ULBP1 to ULBP6) are shown for each of the three monocyte subsets (referred to by R2) based on the published dataset by Monaco et al. Apparently, there is almost no expression of any ULBPs in any of such monocytes apart from an extremely low level of ULBP1 and ULBP3 transcripts in some subsets.

The authors should include proteomic data from the ProteinAtlas dataset to corroborate their conclusions

Reply: Proteomics data of the human ProteinAtlas relate to proteins in the blood (plasma), but to our knowledge there are no proteomics data available for immune cell populations.Since three different databases plus our own analyses show that there are no relevant amounts of RAET1E/ULBP4 transcripts present in blood monocytes, we believe that there is no need to corroborate such data with proteomics data, as protein expression strictly depends on expression of the corresponding transcripts. However, we do show lack of surface binding of the ULBP4-specific mAb DUMO1 corroborating the RNA data.. 

Statistical analysis must be provided.

Reply: We added statistical analyses where meaningful (i.e. Figure 2C). 

Finally, we would like to sincerely thank the Reviewers for all their comments and corrections which substantially improved the revised version of our manuscript.

---

## [Decision Letter · Decision Letter 1]

26 Jan 2021

The NKG2D ligand ULBP4 is not expressed by human monocytes

PONE-D-20-08786R1

Dear Dr. Steinle,

We’re pleased to inform you that your manuscript has been judged scientifically suitable for publication and will be formally accepted for publication once it meets all outstanding technical requirements.

Kind regards,

Silke Appel, PhD (Dr. rer. nat.)

Academic Editor

PLOS ONE

Additional Editor Comments (optional):

Reviewers' comments:

Reviewer's Responses to Questions

**Comments to the Author**

1. If the authors have adequately addressed your comments raised in a previous round of review and you feel that this manuscript is now acceptable for publication, you may indicate that here to bypass the “Comments to the Author” section, enter your conflict of interest statement in the “Confidential to Editor” section, and submit your "Accept" recommendation.

Reviewer #1: All comments have been addressed

Reviewer #2: All comments have been addressed

Reviewer #3: All comments have been addressed

2. Is the manuscript technically sound, and do the data support the conclusions?

Reviewer #1: Yes

Reviewer #2: Yes

Reviewer #3: Yes

3. Has the statistical analysis been performed appropriately and rigorously? 

Reviewer #1: Yes

Reviewer #2: Yes

Reviewer #3: Yes

4. Have the authors made all data underlying the findings in their manuscript fully available?

Reviewer #1: Yes

Reviewer #2: Yes

Reviewer #3: Yes

5. Is the manuscript presented in an intelligible fashion and written in standard English?

Reviewer #1: Yes

Reviewer #2: Yes

Reviewer #3: Yes

6. Review Comments to the Author

Reviewer #1: The authors have addressed all my questions satisfactory and have updated figures and text accordingly. Due to the importance of investigations into specificity of antibodies used in research and their impact on reproducibility and interpretability of research, I recommend this paper for publication.

Reviewer #2: The authors have now addressed the major issues raised in the former manuscript version, and their work has now improved significantly in terms of content and clarity of the message.

Although the technical point relative to NK cell identification strategy (thus relevant for ULBP4 quantification on NK cells shown in Fig 1B rather than quantification of NK cells as a population) was not addressed, I agree with the authors on the focus on the paper being on monocytes rather than on other immune subsets.

Nice work and congratulations to the authors

Reviewer #3: The authors have addressed the questions from the reviewers and the quality of the paper is significantly improved.

7. PLOS authors have the option to publish the peer review history of their article (what does this mean?). If published, this will include your full peer review and any attached files.

Reviewer #1: No

Reviewer #2: No

Reviewer #3: No

---

## [Editor Report · Acceptance letter]

28 Jan 2021

PONE-D-20-08786R1 

The NKG2D ligand ULBP4 is not expressed by human monocytes 

Dear Dr. Steinle:

I'm pleased to inform you that your manuscript has been deemed suitable for publication in PLOS ONE. Congratulations! Your manuscript is now with our production department. 

Kind regards, 

on behalf of

Dr. Silke Appel 

Academic Editor

PLOS ONE